# Bleaching causes loss of disease resistance within the threatened coral species *Acropora cervicornis*

**Erinn M Muller[1]\*, Erich Bartels[2], Iliana B Baums[3]**

[1]Coral Health and Disease Program, Mote Marine Laboratory, Sarasota, United States; [2]Coral Reef Monitoring and Assessment Program, Mote Marine Laboratory, Florida, United States; [3]Department of Biology, Pennsylvania State University, Pennsylvania, United States

**Abstract** Determining the adaptive potential of foundation species, such as reef-building corals, is urgent as the oceans warm and coral populations decline. Theory predicts that corals may adapt to climate change via selection on standing genetic variation. Yet, corals face not only rising temperatures but also novel diseases. We studied the interaction between two major stressors affecting colonies of the threatened coral, *Acropora cervicornis*: white-band disease and high water temperature. We determined that 27% of *A. cervicornis* were disease resistant prior to a thermal anomaly. However, disease resistance was largely lost during a bleaching event because of more compromised coral hosts or increased pathogenic dose/virulence. There was no tradeoff between disease resistance and temperature tolerance; disease susceptibility was independent of *Symbiodinium* strain. The present study shows that susceptibility to temperature stress creates an increased risk in disease-associated mortality, and only rare genets may maintain or gain infectious disease resistance under high temperature. We conclude that *A. cervicornis* populations in the lower Florida Keys harbor few existing genotypes that are resistant to both warming and disease.
DOI: https://doi.org/10.7554/eLife.35066.001

**\*For correspondence:**
emuller@mote.org

**Competing interests:** The authors declare that no competing interests exist.

## Introduction

Genetic diversity within a population leads to varying levels of stress tolerance among individuals (*Sorensen et al., 2001*), and is critical for species survival and persistence in a changing climate (*Hoffmann and Sgrò, 2011*). It is well known that certain corals are more resilient to stress than others, and the genotype of the coral plays a significant role in determining thermal resistance (*Edmunds, 1994*; *Fitt et al., 2009*; *Baird et al., 2009*; *Baums et al., 2013*; *Kenkel et al., 2013*), with a heritable component (*Kenkel et al., 2013*; *Dixon et al., 2015*; *Polato et al., 2013*). Tolerance to stress may also be a result of different symbiotic algal species (*Symbiodinium* spp.) or even of different genotypes (i.e. strains) of certain *Symbiodinium* species that reside within the coral host (*Grégoire et al., 2017*; *Parkinson and Baums, 2014*; *Fabricius et al., 2004*). Furthermore, additional threats to corals, such as infectious disease outbreaks and ocean acidification, are affecting populations in combination with temperature anomalies (*Hoegh-Guldberg et al., 2007*). Evidence suggests that many species possess the ability to produce broad-spectrum defense mechanisms (*de Nadal et al., 2011*). Similarly, coral populations showing resilience to high water temperatures constitutively frontload the expression of genes related to heat tolerance in concert with several genes influencing the host innate immune response (*Barshis et al., 2013*), suggesting these corals may also have evolved the general ability to tolerate a multitude of threats. A critical question is whether certain coral genotypes, existing within the same environment, are generally more stress

**eLife digest** The staghorn coral was once prevalent throughout the Florida Reef Tract. However, the last few decades have seen a substantial reduction in the coral population because of disease outbreaks and increasing ocean temperatures. The staghorn coral shows no evidence of natural recovery, and so has been the focus of restoration efforts throughout much of the Florida region.

Why put the time and effort into growing corals that are unlikely to survive within environmental conditions that continue to deteriorate? One reason is that the genetic make-up – the genotype – of some corals makes them more resilient to certain threats. However, there could be tradeoffs associated with these resilient traits. For example, a coral may be able to tolerate heat, but may easily succumb to disease.

Previous studies have identified some staghorn coral genotypes that are resistant to an infection called white-band disease. The influence of high water temperatures on the ability of the coral to resist this disease was not known. There also remained the possibility that more varieties of coral might show similar disease resistance.

To investigate Muller et al. conducted two experiments exposing staghorn coral genotypes to white-band diseased tissue before and during a coral bleaching event. Approximately 25% of the population of staghorn tested was resistant to white-band disease before the bleaching event. When the corals were exposed to white-band disease during bleaching, twice as much of the coral died.

Two out of the 15, or 13%, of the coral genotypes tested were resistant to the disease even while bleached. Additionally, the level of bleaching within the coral genotypes was not related to how easily they developed white-band disease, suggesting that there are no direct tradeoffs between heat tolerance and disease resistance. These results suggest that there are very hardy corals, created by nature, already in existence. Incorporating these traits thoughtfully into coral restoration plans may increase the likelihood of population-based recovery.

The Florida Reef Tract is estimated to be worth over six billion dollars to the state economy, providing over 70,000 jobs and attracting millions of tourists into Florida each year. However, much of these ecosystem services will be lost if living coral is not restored within the reef tract. The results presented by Muller et al. emphasize the need for maintaining high genetic diversity while increasing resiliency when restoring coral. They also emphasize that disease resistant corals, even when bleached, already exist and may be an integral part of the recovery of Florida's reef tract.

DOI: https://doi.org/10.7554/eLife.35066.002

resistant compared with other conspecifics. And additionally, does stress resistance in one trait predict stress resistance in another?

The Caribbean coral species, *Acropora cervicornis,* was one of the most common corals within the shallow reefs of the Western Atlantic and Caribbean several decades ago (*Pandolfi, 2002*). However, over the last 40 years multiple stressors including infectious disease, high sea surface temperatures, overfishing and habitat degradation have caused a 95% population reduction (*Acropora Biological Review Team, 2005*). *A. cervicornis* is now listed as threatened under the U.S. Endangered Species Act. Significant loss of Caribbean *Acropora* species was attributed to white-band disease outbreaks that spread throughout the region in the late 1970' s and early 1980' s (*Aronson and Precht, 2001*). While the disease-causing agent of white band has not been identified, the pathogen is likely bacterial in nature (*Peters, 1984*; *Kline and Vollmer, 2011*; *Sweet et al., 2014*). This disease continues to cause mortality across populations (*Miller et al., 2014*) and especially Florida (*Precht et al., 2016*). Recently, variability of *Acropora* spp. susceptibility to disease has been explored and documented. For example, 6% of the *A. cervicornis* tested in Panama were resistant to white-band disease (*Vollmer and Kline, 2008*). Additionally, long term monitoring of *A. palmata* in the US Virgin Islands indicated that 6% of 48 known genets showed no disease signs over eight years; perhaps indicating a small disease resistant population (*Rogers and Muller, 2012*). Hence, there are disease resistant variants within some locations, although they may be low in abundance. Regardless, anomalously high water temperatures are increasing the probability of disease

occurrence throughout the Caribbean (*Muller et al., 2008*; *Miller et al., 2009*; *Randall and van Woesik, 2015*) and field monitoring suggests that bleached corals are more susceptible to disease (*Muller et al., 2008*). Alternatively, recent field-based observations suggest that there is a negative association between heat tolerance and disease susceptibility in *A. cervicornis* (*Merselis et al., 2018*). Here, we determine whether high water temperature changes the susceptibility of disease resistant variants and explore the potential relationship between disease resistance and susceptibility to high temperatures.

Tropical reef-building corals gain a majority of their carbon from their algal symbionts (*Muscatine et al., 1984*) and thus the stress response of the coral animal has to be viewed in the context of its symbiotic partner or partners. Prolonged temperature stress causes the disassociation between the coral host and the single-celled algae (*Symbiodinium* spp) residing within its tissues, a phenomenon called bleaching. Studies of the immune protein concentrations within corals indicate a suppressed immune system when corals bleached (*Mydlarz et al., 2009*). Furthermore, immune-related host gene activity is suppressed for at least a year after bleaching occurs, at least for some species (*Pinzón et al., 2015*). While some coral species harbor multiple *Symbiodinium* species in the same colony or over environmental gradients, others associate with only one *Symbiodinium* species (*LaJeunesse, 2002*). *Symbiodinium* species differ in their heat tolerance (*Berkelmans and van Oppen, 2006*), however, evidence of an association between *Symbiodinium* species identity and coral host disease susceptibility, is not well studied and equivocal (*Correa et al., 2009*; *Rouzé et al., 2016*). *Acropora cervicornis* can harbor several species of *Symbiodinium* (*Baums et al., 2010*), but it is often dominated by *Symbiodinium 'fitti' (nominem nudum)*. No study has addressed whether different strains of the same *Symbiodinium* species influence infectious disease susceptibility in the coral host. Yet, different strains of a single *Symbiodinium* species can affect coral physiology (*Howells et al., 2011*) and thus we aimed to determine the influence of *Symbiodinium* strain diversity on coral disease susceptibility and bleaching in *A. cervicornis*.

Coral nurseries provide a unique opportunity to test the effect of multiple stressors on coral survival and adaptation in a common garden environment. Nurseries propagate colonies via asexual fragmentation providing experimental replicates for each host genotype/*S. 'fitti'* combination. Host genotypes display differences in growth, linear extension, and thermal tolerance, which are maintained within the common garden environment (*Lohr and Patterson, 2017*). Here, we use 15 common garden-reared host genotypes infected with known *S.' fitti'* strains. Genets were exposed to white-band disease homogenates under control conditions and then again after a period of elevated water temperatures. We measured the rate of infection and the performance of the symbiosis under control and treatment conditions to evaluate the hypothesis that infection resistance predicts bleaching resistance in the holobionts. The objectives of the present study were to (1) determine the relative abundance of genotypes of *Acropora cervicornis* from the lower Florida Keys that were resistant to disease, (2) characterize the *Symbiodinium* strains within each host and explore the potential relationship between the algal symbionts and disease susceptibility, and (3) quantify the relative change in disease risk when corals were bleached.

## Results

### Photochemical efficiency

The photochemical yield ($F_v/F_m$) of all fragments, prior to visual bleaching in August 2015, averaged 0.457 (± .015 SE). However, by September 2015, colonies in the nursery had visually bleached after experiencing temperatures ~2°C above historical averages (*Figure 1*), represented by 8 degree heating weeks under NOAA's Coral Reef Watch products. By this time, all of the corals had visibly turned white and the photochemical yield of the corals dropped to 0.148 (± 0.008 SE). While there was a gradual reduction in photochemical yield from August to September, fragments in the first three pre-bleaching trials were significantly higher than the post-bleaching trial (*Figure 2A*, *Supplementary file 1*, *Figure 2—figure supplement 1*), as expected. In the August pre-bleaching trials, there were significant differences among the photochemical efficiency of *S. fitti* associated with different host genets ($X^2 = 51.173$, df = 14, p<0.001, *Figure 2A*). Photochemical efficiency also differed among *S. fitti* associated with different host genets after bleaching ($X^2 = 24.42$, df = 14,

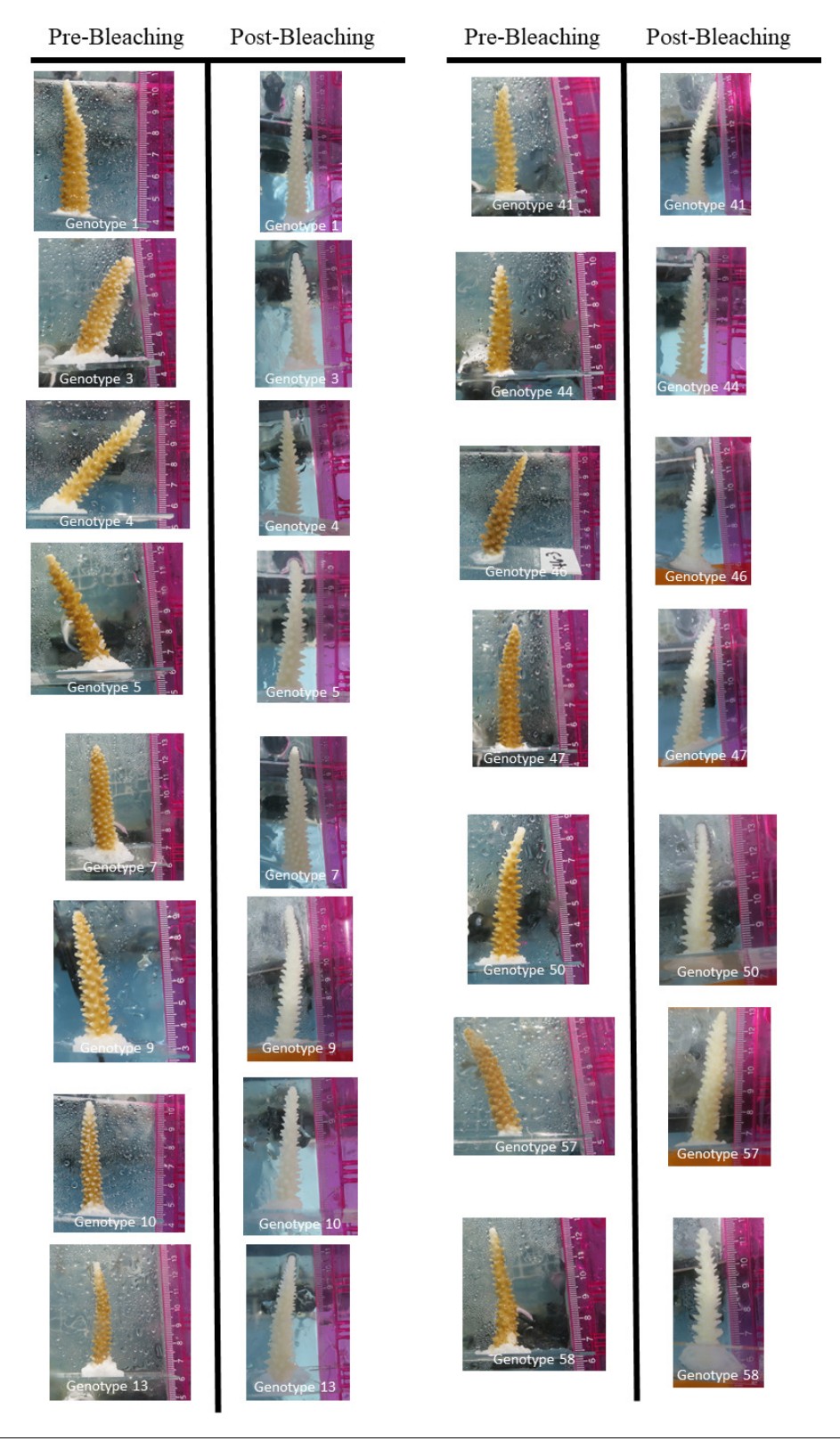

**Figure 1.** Digital photographs of each genotype of *Acropora cervicornis* used within the present study. Photographs labeled pre-bleaching were taken in August 2015 and those labeled post-bleaching were taken in September 2015.

DOI: https://doi.org/10.7554/eLife.35066.003

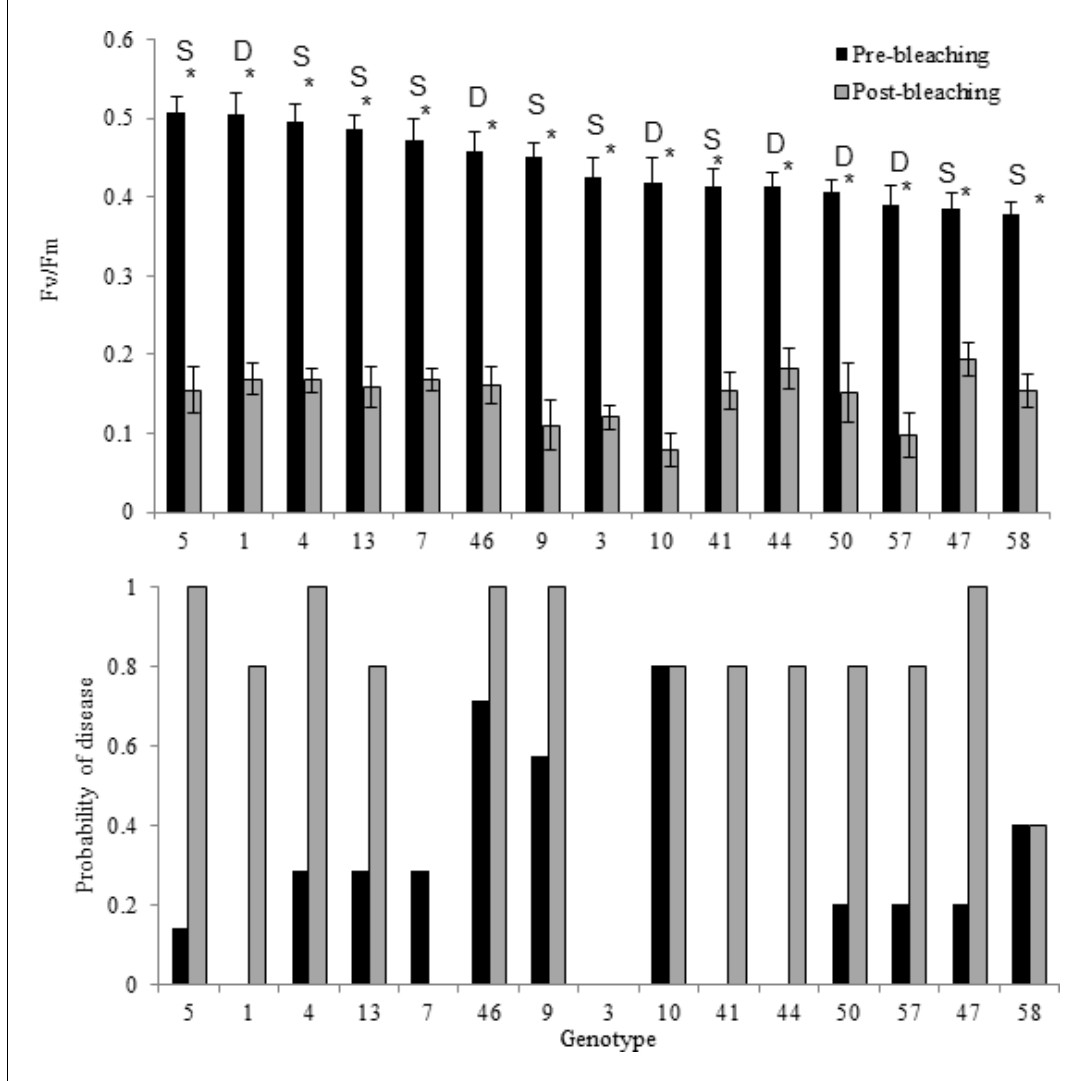

**Figure 2.** Comparison between photochemical yield and disease susceptibility measurements taken pre-bleaching (August) and post-bleaching (September) for 15 genotypes of *Acropora cervicornis*. (**A**) Average photochemical yield ($F_v/F_m$) of *S. fitti* associated with 15 different genets of *Acropora cerviconis* after dark acclimation occurred. Measurements were taken in August, prior to bleaching (black bars) and in September, post bleaching (grey bars). Pre- and post-bleaching photochemical yields were significantly different (asterisk). Labels above the- bars represent the *Symbiodinium fitti* strain where S indicates a F421 strain and D represents all other strains detected. Error bars represent the standard error of the mean. (**B**) The relative percent of ramets that showed disease signs within each of the 15 genets of *Acropora cervicornis* exposed to a disease homogenate (n = 5–7 ramets per genet per treatment). Bars represent disease susceptibility for each genet in August, prior to bleaching (black) or in September, post bleaching (grey).

DOI: https://doi.org/10.7554/eLife.35066.004

The following figure supplements are available for figure 2:

**Figure supplement 1.** Photochemical efficiency ($F_v/F_m$) of all genotypes tested within each trial in August/early September (Trials 1–3, pre-bleaching), and in September (trial 4, post-bleaching).
DOI: https://doi.org/10.7554/eLife.35066.005

**Figure supplement 2.** Average disease susceptibility of corals that contain either F421 strain of *symbiodinium* (Single, n = 11) or other strains of *Symbiodinium* (Diverse, n = 4).
DOI: https://doi.org/10.7554/eLife.35066.006

**Figure supplement 3.** Average disease susceptibility of corals that contain either F421 strain of *Symbiodinium* (Single, n = 11) or other strains of *Symbiodinium* (Diverse, n = 4) over the different trials within the study.
DOI: https://doi.org/10.7554/eLife.35066.007

p=0.04), although the relative pattern among *S. fitti* associated with different host genets changed (*Figure 2A*).

## Disease susceptibility

A total of 25 out of the 75 fragments exposed to the disease homogenate showed signs of white-band-disease-associated mortality within the first seven days after exposure during the August, pre-bleaching trials. Only one fragment (genet 46), out of the 75 total fragments showed signs of disease in the control treatment, within the experimental period. There was high variation in disease susceptibility among genets, with susceptibility values ranging from 0% to 80% (*Figure 2B*). Four genets showed complete resistance to the disease homogenate, with no replicate fragments showing any signs of tissue loss after disease exposure. The median susceptibility value among the different genets was 20%.

Results differed when the same genets were exposed to the disease homogenate after they were bleached. A total of 55 out of 75 fragments lost tissue after exposure to the disease homogenate. Additionally, 13 out of the 75 control fragments died when bleached. Values ranged from 100% susceptibility to disease-induced mortality within five genets, to one genet that maintained disease tolerance, even when bleached (genet 3; *Figure 2B*). Median susceptibility was 80% among the genets when the corals were bleached. The generalized linear model, which tested whether maximum quantum yield affected disease presence or absence for each replicate genotype indicated there was no significant effect of the photochemical yield on disease susceptibility, even within corals exposed to the disease homogenate, during either the pre-bleaching (z = 0.132, p=0.895) or post-bleaching experiments (z = −1.579, p=0.114). Also, there was no effect of the average change in photochemical yield within each genotype on disease susceptibility (pre-bleaching: z = 0.555, p=0.579; post-bleaching: z = −0.023, p=0.982).

The mixed-effect generalized linear model showed that the treatment effect was significant within both the pre-bleaching (z = 2.263, p=0.0234) and post bleaching trials (z = 3.515, p<0.001), with higher disease presence within corals exposed to the disease homogenate. However, there were no significant differences detected among genotypes within trials (pre-bleaching: z = 0.416, p=0.677; post-bleaching: z = −0.243, p=0.808), nor was there a significant interaction between treatment and genotype within each trial (pre-bleaching: z = 0.090, p=0.928; post-bleaching: z = 0.697, p=0.486). There was also no difference in disease presence or absence among the three trials that created the pre-bleaching experiment (z = −1.308, p=0.191), suggesting that pooling data from these three trials was appropriate.

## *Symbiodinium fitti* strains

A total of six different *S. fitti* strains were found within the 15 coral host genets (*Supplementary file 2*). Note that no other *Symbiodinium* clades have been detected in the Mote *in situ* nursery *A. cervicornis* fragments above background levels (*Parkinson et al., 2018a*) or in other offshore *A. cervicornis* colonies in the Keys (*Baums et al., 2010*). A majority of the host genets tested (11/15) harbored a single *S fitti* strain consistently through time (strain F421; *Figure 2A*). The other four *Symbiodinium* strains were associated with a single coral genet each (See *Supplementary file 2*).

There was no significant difference in photochemical efficiency between corals harboring the common F421 *Symbiodinium* strain and those that harbored the other unique strains, pre- and post-bleaching (pre-bleaching yield: t = −0.13, df = 13, p=0.99; post-bleaching yield: t = −0.244, df = 13, p=0.811). Similarly, there was no influence of *Symbiodinium* strain on the amount of change in photochemical efficiency between the pre- and post-bleaching experiments (change in yield: t = 0.172, df = 13, p=0.866).

There was no significant difference in disease susceptibility when corals contained the single *Symbiodinium* strain F421 compared with the other corals that hosted different *Symbiodinium* strains (pre-bleaching disease: $X^2$ = 0.039, df = 1, p=0.842, post-bleaching disease: $X^2$ = 0.079, df = 1, p=0.779; *Figure 2—figure supplement 2*). This trend held through time (*Figure 2—figure supplement 3*). Among the eleven coral genets that harbored *S. 'fitti'* F421, disease susceptibility ranged from 0% (genet 3) to 70% (genet 46) during pre-bleaching trials (*Figure 2B*). *S. fitti* strain identity also did not influence disease susceptibility for the post bleaching exposures (*Figure 2B*).

## Relative risk analyses

Under pre-bleaching conditions, the tested *A. cervicornis* genets were almost three times as likely to experience disease-induced mortality when exposed to the disease homogenate over healthy homogenates (median Bayesian relative risk = 2.77, *Figure 3A*). The relative risk analysis also showed evident differences among genets, even though the frequenstist statistics did not indicate significant differences among genets within the generalized linear models. For example, genets 1, 3, 41 and 44 showed no increase in disease-induced mortality after disease exposure, with median relative risk values of these four genets near 1, that is they were resistant. The 11 other genets, however, showed an increase in disease risk after exposure to the disease homogenate (i.e. they were susceptible), with statistically significant increased relative risk values for genets 9 and 10 (*Figure 3A*, *Supplementary file 3*).

Post-bleaching, the overall likelihood of disease-induced mortality increased by about three-fold (median Bayesian relative risk = 3.33, *Figure 3B*) when the corals were bleached and exposed to the disease homogenate, compared with corals that were bleached and exposed to the healthy homogenate. Again, substantial variation among genets was detected, although there were now six genets that showed a significant increase in disease risk (genets 5, 41, 44, 46, 47, and 50; *Figure 3B*, *Supplementary file 4*). Only one genet, genet 7, was affected by disease pre-bleaching, but not post-bleaching, whereas genet 3 showed apparent complete immunity whether exposed to a disease homogenate when bleached or not.

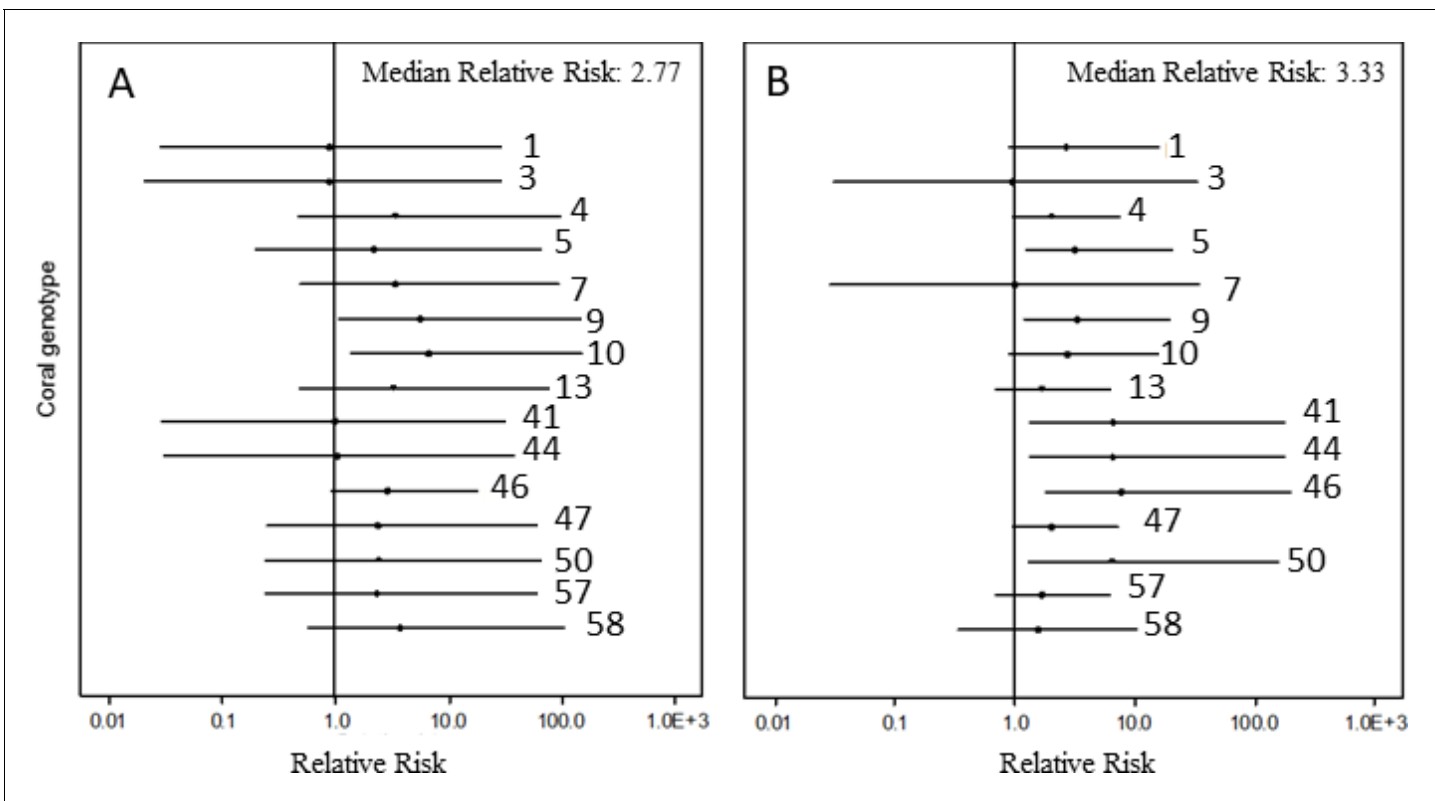

**Figure 3.** Caterpillar plot of the Bayesian relative risk analysis of *Acropora cervicornis* on the log scale. (A) relative risk increase after exposure of corals to the disease homogenate compared with corals that were exposed to the healthy homogenate under non-bleaching conditions and (B) relative risk increase after exposure of corals to the disease homogenate when corals were bleached compared with corals that were bleached and exposed to the healthy homogenate. Dots represent the median risk value of that genet, lines depict the 95% credible interval of the Bayesian analysis. Credible intervals entirely above (below) a relative risk of 1 indicate a significant increase (decrease) in disease risk after exposure to the risk. Credible intervals that include a value of 1 indicate no significant influence of exposure to the risk.
DOI: https://doi.org/10.7554/eLife.35066.008

## Bacterial communities of homogenates

There were no differences detected in the bacterial community of the homogenates among trials ($F_{(1,7)}$=1.42, p = 0.219), nor did a single operational taxonomic unit significantly differ in relative abundance among trials (1,253 OTUs tested using nonparametric Kruskal Wallis tests). The PERMA-NOVA analysis also showed no statistical difference between the bacterial OTU community of the healthy homogenate and the disease homogenate ($F_{(1,6)}$ = 1.962, p = 0.134), which was primarily because one healthy homogenate sample was similar to the disease homogenate samples (*Figure 4*). Comparisons of the relative abundances of the major bacterial classes showed no significant differences between the healthy and the disease homogenates using nonparametric tests. However, when the one outlier sample was removed there was a significantly higher abundance of Actinobacteria ($X^2$ = 4.5, df = 1, p = 0.034) and significantly lower abundance of Alphaproteobacteria ($X^2$ = 4.5, df = 1, p = 0.034) within the healthy samples compared with the diseased samples (*Figure 4—figure supplement 1*).

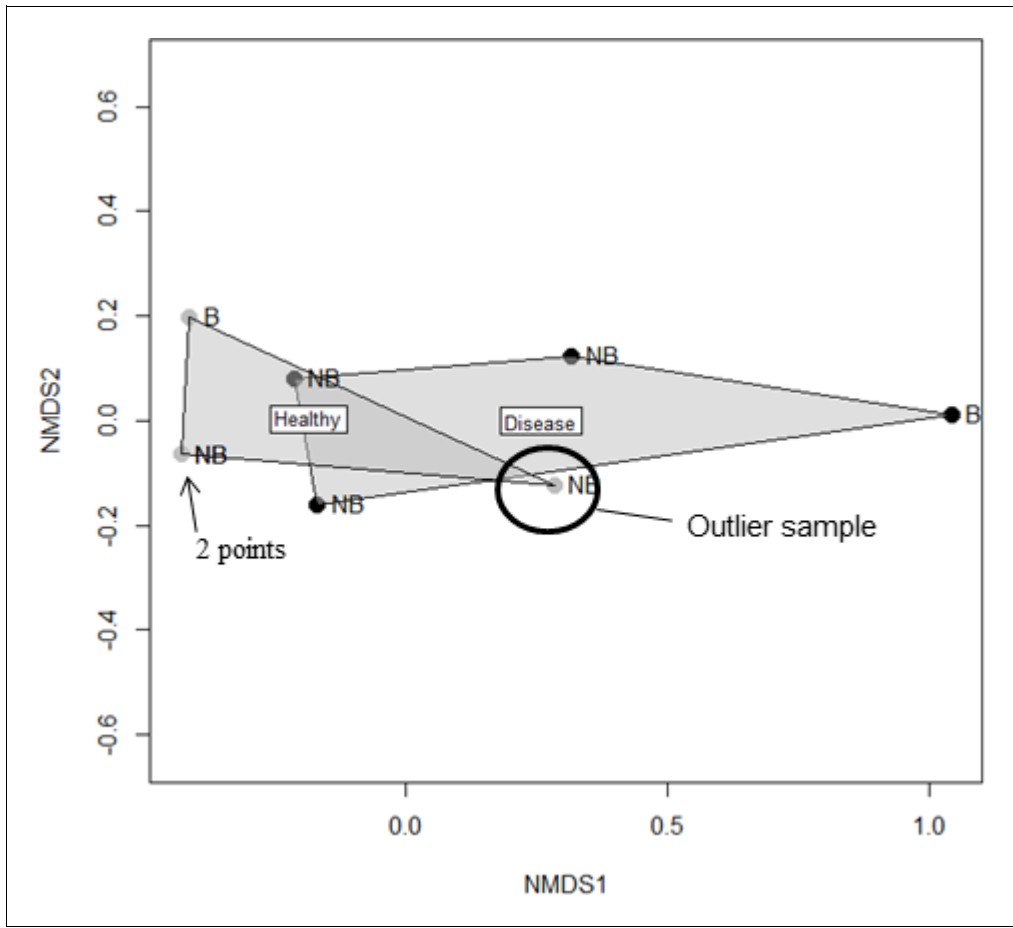

**Figure 4.** A nonmetric multidimensional scaling plot of the bacterial community within healthy and diseased tissue homogenates after Illumina sequencing of the 16S rRNA gene. Grey dots represent the healthy homogenate samples and the black dots represent the disease homogenate samples. NB denotes the samples that came from corals in August 2015, when they were not bleached. B denotes the samples that came from corals in September 2015, when corals were bleached. Note that two healthy points were so similar in multidimensional space that they overlap within the present figure.

DOI: https://doi.org/10.7554/eLife.35066.009

The following figure supplement is available for figure 4:

**Figure supplement 1.** Histogram showing the average relative abundance of each major bacterial class identified within the healthy and disease tissue homogenates, excluding the one outlier sample.

DOI: https://doi.org/10.7554/eLife.35066.010

## Discussion

Our results suggest that disease resistance and temperature tolerance evolve independently within *A. cervicornis* of the lower Florida Keys. Some genets showed significantly lower levels of bleaching compared with others, but had varying levels of disease susceptibility. These temperature tolerance and disease resistance traits were driven by the host genotype rather than strain variation in *Symbiodinium 'fitti'*, the dominant symbiont in the coral colonies (*Baums et al., 2010*; *Parkinson et al., 2018b*). In the US Virgin Islands, bleached state, rather than temperature, influenced *A. palmata* susceptibility to disease; and bleaching resistance conferred disease resistance (*Muller et al., 2008*). Within the present study, however, none of the Florida coral genets appeared to be resistant to bleaching, which may have contributed to high rates of disease risk after bleaching occurred. Variability in disease susceptibility was also reduced as several bleached genets with moderate to low levels of infectious disease susceptibility pre-bleaching showed high mortality levels when exposed to disease. The almost complete loss of white-band disease resistance after temperature-induced bleaching in *A. cervicornis* suggests that current adaptations to disease infection provide only limited protection against future colony mortality in light of rising ocean temperatures.

Without knowing the primary pathogen of white-band disease, it is difficult to definitively differentiate the influence of the coral host state and the pathogenic dose between the pre and post-bleaching experiments. Although the physiological state of the host genotype may have contributed to the higher risk of disease when bleached, the potential pathogenic dose or virulence could have changed between the August (pre-bleaching) and September (post-bleaching) trials as well. Increased water temperatures can lead to higher growth rates of bacterial pathogens (*Remily and Richardson, 2006*) and also lead to increased virulence (*Kushmaro et al., 1998*; *Toren et al., 1998*; *Harvell et al., 2002*; *Ben-Haim et al., 2003*). We did not detect a difference in the bacterial community of the homogenates among trials, only among treatments. However, without an identified primary pathogen, it was impossible to know whether pathogenic virulence could have influenced the results. Regardless of the mechanism, higher disease risk is evident when corals experience thermal anomalies and increased disease prevalence is likely as the world's climate continues to warm.

Disease resistance itself, was evident within both the pre and post-bleaching experiments. Prior to bleaching, four out of the 15 genets, or 27%, of the tested population showed complete resistance to disease exposure. In comparison, Panama and the USVI harbored only approximately 6%, of disease resistant genets (*Vollmer and Kline, 2008*; *Rogers and Muller, 2012*). Interestingly, two genets showed resistance after bleaching occurred, one of which was also disease resistant prior to bleaching (genotype 3). Disease resistance could be provided by a certain gene or set of genes within the host genome (*Libro and Vollmer, 2016*), a unique microbiome within the tissue or mucus of the disease resistant corals (*Gignoux-Wolfsohn et al., 2017*), or could be influence by the energy reserves within these particular coral genotypes. For example, each genotype of *A. cervicornis* interacts with their environment eliciting differential growth rates, bleaching susceptibility and recovery from bleaching (*Drury et al., 2017*). Subsequent studies will focus on identifying the mechanism driving disease resistance within the *A. cervicornis* corals used in the present experiments, with the recognition that a combination of these potential pathways is also possible.

Resistance to disease in *Acropora cervicornis* populations from Panama may be driven by constitutive gene expression (*Libro and Vollmer, 2016*). Particularly, genes involved in RNA interference-mediated gene silencing are up-regulated in disease resistant corals, whereas heat shock proteins (HSPs) were down-regulated. Libro and Vollmer (*Libro and Vollmer, 2016*) postulated that reduced HSPs in disease resistant corals from Panama may indicate high temperature resistance. In the Florida population studied here, however, three coral genotypes that were resistant to disease prior to bleaching showed a similar level of bleaching susceptibility to those that were susceptible to disease. This indicates that there was no obvious tradeoff or shared protection between disease resistance and temperature resistance for *A. cervicornis* within the lower Florida Keys. Gene flow among *A. cervicornis* populations is limited (*Baums et al., 2010*; *Vollmer and Palumbi, 2007*; *Hemond and Vollmer, 2010*) thus spatial variation in genetic trait architecture is possible and would complicate predictions of how corals may adapt to climate change (*Bay et al., 2017*).

A higher occurrence of disease resistance of *Acropora* within the Florida Keys (27%) compared with populations tested in Panama and the USVI (6 and 8% respectively) may be a result of more intense selection events within Florida compared with other locations in the Caribbean. However,

there may also be methodological differences among studies that make direct comparisons challenging. Since nursery corals originate from fragments of opportunity within the wild population, the corals used in this study should represent a random subset of the wild population. The density and overall abundance of wild colonies of *Acropora* spp. within the lower Florida Keys has continued to decline (*Patterson et al., 2002*). Significant direct anthropogenic impacts in the Florida Keys (*Lapointe et al., 2004*; *Sutherland et al., 2011*), disease outbreaks (*Aronson and Precht, 2001*; *Patterson et al., 2002*), as well as several recent bleaching events (*Manzello, 2015*; *Lewis et al., 2017*), may have fostered the persistence of only extremely hardy coral genets. The documentation of over a quarter of the tested population showing signs of disease resistance under non-bleaching conditions provides a glimmer of hope that natural evolutionary processes may allow for the persistence of a population in peril, such as *Acropora cervicornis*. Future work should concentrate on determining the degree of spatial variability among temperature and infectious disease resistance traits and their interactions.

Disease susceptibility of *A. cervicornis* was more strongly linked to the coral host genet, rather than the algal symbiont strain. The high variability in disease susceptibility when host genets associated with strain F421 suggests that although *Symbiodinium* strain can influence phenotypic physiology of the host (*Grégoire et al., 2017*; *Parkinson and Baums, 2014*; *Parkinson et al., 2015*), white-band disease resistance is likely related to *A. cervicornis* host genotype. Furthermore, corals with the common F421 strain showed similar levels of disease susceptibility, both before and after bleaching, compared with corals that hosted all other strains. Future research should aim to evaluate the influence of additional *Symbiodinium fitti* strains on host disease resistance, to further evaluate the hypothesis that diversity of *Symbiodinium* strains within the population has little direct influence on holobiont disease susceptibility. Previous research showed *Symbiodinium* clade had no influence on disease susceptibility of several diseases infecting various coral species in the Atlantic and Caribbean region (*Correa et al., 2009*). The present results suggest that Correa et al's conclusions may extend from the algal clade (genus level) to the *Symbiodinium* strain (within species level).

The dire state of coral populations, such as *A. cervicornis*, has forced a more interventionist approach to coral conservation because natural population recovery may not be possible on some reefs that are lacking sources of new recruits. Selective breeding of stress resistant host genotypes, experimental evolution of stress-resistant symbiont cultures, and gene therapy are now all being considered (*Mascarelli, 2014*; *van Oppen et al., 2015*; *van Oppen et al., 2017*). Design of effective breeding strategies for hosts and symbionts, however, requires knowledge of how genotypes respond to interacting stressors, not just temperature increases alone. Of particular interest, was the discovery of a genet that became disease tolerant when bleached (genet 7), although further testing of this genotype should occur to validate the results of the present study, which had limited replication. These results have important implications for selective breeding initiatives. For example, genet 7 would not have been a prime candidate for selective breeding based on disease resistance or bleaching susceptibility alone. Yet its apparent gain of disease resistance after bleaching makes it a potentially valuable genotype. If selective breeding initiatives focus on resistance to single stressors, interactive phenotypes, such as increased disease resistance under bleaching conditions, may be lost from the population. The consequences of these choices may be unpredictable and risky.

In conclusion, under non-stressful conditions, disease resistance within the lower Florida Keys *A. cervicornis* population appears relatively prevalent compared to other regions in the Caribbean, perhaps because of many previous natural selection events over the last several decades. Disease outbreaks within *Acropora* spp. began in the late 1970 s and early 1980 s and continues to occur on contemporary reefs (*Aronson and Precht, 2001*; *Miller et al., 2014*; *Rogers and Muller, 2012*; *Patterson et al., 2002*). Resistance to high water temperature anomalies, however, appears decoupled from disease resistance as all genets appeared visibly bleached and showed significant loss of photochemical efficiency during the 2015 bleaching event. Historical records from 1870 to 2007 indicate that Florida, and most of the Caribbean, have had substantially longer return periods between potential bleaching events compared with temperature anomalies observed over the last decade, thus selection strength for thermal tolerance may have been small prior to recent years (*Thompson and van Woesik, 2009*). The bleaching event increased the risk of mortality from disease, whether it was from higher disease susceptibility or increased pathogenic load and/or virulence, and caused almost all previously resistant corals to become disease susceptible. Importantly, these results suggest that there is no tradeoff or shared protection between disease resistance and

temperature tolerance within *Acropora cervicornis* of the lower Florida Keys. The present study shows that susceptibility to temperature stress creates an increased risk in disease-associated mortality, and only rare genets may maintain or gain infectious disease resistance under high temperature. We conclude that *A. cervicornis* populations in the lower Florida Keys harbor few existing genotypes that are resistant to both warming temperatures and infectious disease outbreaks and that recurring warming events may cause continued loss of disease resistant genotypes.

## Materials and methods

### Experimental design

A total of 10–12 replicate fragments (ramets) each from 15 genotypically distinct host colonies (genets), as determined via microsatellite genotyping (see below), were collected from the Mote Marine Laboratory *in situ* coral nursery in August, 2015. Number of replicates and genotypes were determined based on the maximum number available within the nursery for experimentation with considerations of additional spatial constraints within the wetlab area. Genets were originally collected from nearby reefs (<20 km maximum linear distance) and had been growing in the nursery for at least 5 years (*Supplementary file 5*). The small spatial scale over which genets were originally sourced suggests that these belong to the same population (*Baums et al., 2010*; *Drury et al., 2016*). Each ramet was cut from the donor colony using metal pliers and was approximately 5 cm in length. Ramets were transported in ambient seawater to Mote Marine Laboratory. Corals were mounted on PVC pipe plugs or glass slides using cyanoacrylate gel (Bulk Reef Supply extra thick super glue gel). Permit restrictions meant that three disease exposure trials had to be conducted rather than testing all genotypes at once. The total number of genets varied for each trial, but each of the 15 genets was comparatively represented by the conclusion of the experiments. The sum of the three trials resulted in a total of 5–7 ramets per treatment (disease vs control) of each of the 15 different genets; a total of 170 corals (see *Supplementary file 6* for details). For each trial, ten aquaria were held within a single raceway, which contained a recirculating water bath kept at approximately 25°C. One ramet of one genet was placed within a 19 L glass aquarium that contained 9.5 L of seawater, thus each aquarium contained a single ramet of each genet for each trial with a maximum amount of 15 corals per tank. Water flow within the aquaria was maintained using 340 L per hour submersible powerheads. Temperature, salinity and pH were measured daily to ensure consistency among tanks. Corals were allowed to acclimate to tank conditions for 3 days prior to disease exposure. During the acclimation period the photochemical efficiency of the corals was measured using an Imaging Pulse Amplitude Modulation fluorometer (IPAM Walz, Germany). Measurements were taken at least 1 hr after sunset. PAM fluorometry is a useful tool for quantifying the physiological parameters of the symbiotic algae found within scleractinian corals. Peak photochemical efficiency typical yields values between 0.5 and 0.7, depending on the species, whereas reduced values indicate photochemical inhibition (*Fitt et al., 2001*). Within the present study, photochemical efficiency ($F_v/F_m$) was used as a proxy for coral bleaching. Although $F_v/F_m$ was not a direct measurement of bleaching, visual qualitative assessment showed that each genotype was regularly colored during the August trials (*Figure 1*).

### Disease and healthy homogenates

After the acclimation period, five randomly selected tanks were treated with a disease tissue homogenate, whereas the remaining five tanks were treated with a healthy tissue homogenate using a modified protocol developed by Vollmer and Kline (*Vollmer and Kline, 2008*). To create the disease homogenate, fragments of *A. cervicornis* showing signs of active white-band disease were collected from an offshore reef at approximately 7.6 m depth (24.54129° N, 81.44066° W). Live tissue from diseased fragments was removed by airbrushing off the tissue within 5 cm of the advancing band using 0.2 micron filter-sterilized seawater. To increase the likelihood that the disease homogenate contained active and viable pathogens, the homogenate from several different diseased corals was pooled into one sample. Surface area of diseased tissue acquired to create the slurry was approximately 10 cm$^2$ per fragment, equating to ~11 cm$^2$ of coral tissue per 100 ml of slurry. Approximately 100 ml of the disease homogenate was poured into each of the five treatment tanks.

*Acropora cervicornis* fragments from the Mote Marine Laboratory *in situ* coral nursery were collected to create the healthy tissue homogenate; reducing impacts to the wild population of *Acropora cervicornis*. The healthy tissue of 11 fragments, all approximately 5 cm in length, was airbrushed using filter-sterilized seawater and collected in 50 ml plastic tubes. Surface area of each healthy fragment was approximately 10 cm², equating to ~11 cm² of coral tissue per 100 ml of slurry. Approximately 100 ml of the healthy tissue homogenate was poured into each of the five control tanks. This procedure was repeated for each of the three August trials. All experimental corals were abraded near the base of the fragment prior to treatment using a sterile scalpel to increase the probability of disease infection.

## Exposure during bleaching

In September, 2015 another set of 10 ramets from the same 15 genets of *Acropora cervicornis* were collected from the Mote Marine Laboratory *in situ* coral nursery. By this time, the nursery corals had been experiencing anomalously high water temperatures reaching approximately ~2°C above historical averages, represented by 8 degree heating weeks under NOAA's Coral Reef Watch products (www.coralreefwatch.noaa.gov) (*Strong et al., 2011*). Corals were collected and mounted similarly to the August collection and allowed to acclimate for three days in tanks at 27.5°C. During that time the $F_v/F_m$ of each coral was determined using the IPAM. Visual qualitative assessment showed that each genotype was completely white at the onset of the September trial (*Figure 1*). Fresh samples of diseased corals were collected from the same reef area as the August experiment, although from different colonies because the original diseased colonies had died. The healthy homogenate was again created from nursery corals that showed no apparent signs of tissue loss.

Infectious dose between the pre- and post-bleaching trials was standardized in several ways even though the primary pathogen of white-band disease is unknown. First, the disease samples were collected from the same area (~100 × 100 m in reef area) for each disease trial. Second, the area of diseased tissue used to create the slurry was standardized for the healthy and the disease slurries, and was also consistent among trials. Third, the tissue was collected within a standardized distance away from the disease margin to create the disease slurry. Fourth, disease samples collected in the field showed similar rates of tissue loss within the host colony. However, because the primary pathogen of white-band disease is still unknown, the dose of the infectious agent could not be determined within each slurry combination. This limits the ability to compare results between each experiment, but does not inhibit interpretation of the results within each experiment. Samples of each homogenate were processed for 16S rDNA in an effort to characterize the bacterial community of each trial (see Materials and methods below).

## Infection rates

For both the August and September experiment, corals were monitored twice a day, in the morning and early evening hours, for seven days post treatment. Signs of disease mortality were recorded when observed and photographs were taken with a ruler. A new infection was defined as recently exposed skeleton caused by tissue sloughing off, often occurring from the base of the fragment and progressing towards the branch tip (*Figure 5*). Because the corals were already white during the post-bleaching experiment, the visual sign of mortality was determined by the apparent loss of tissue and simultaneous accumulation of algae on the coral skeleton. During bleaching, this often occurred over the entire fragment rather than a visual progression from the base to the tip. Mortality within the controls often showed the same signs of tissue loss for both the pre-bleaching and post-bleaching experiment. The number of ramets per genet that showed signs of disease mortality was used as the risk input within the relative risk analysis (see below). In this regard, the proportion of ramets of a given genet that showed disease signs reflected the level of disease resistance for that genet. All data are fully available through the Biological and Chemical Oceanography Data Management Office, an open access data repository (http://www.bco-dmo.org/dataset/642860).

## Statistical analyses

We used the Two Sample Welch's T test or Kruskal Wallis tests to determine whether the photochemical efficiency of each coral genet changed between the pre and post-bleaching experiments, depending on the data set passing parametric assumptions. A Kruskal Wallis test with a Dunn's post

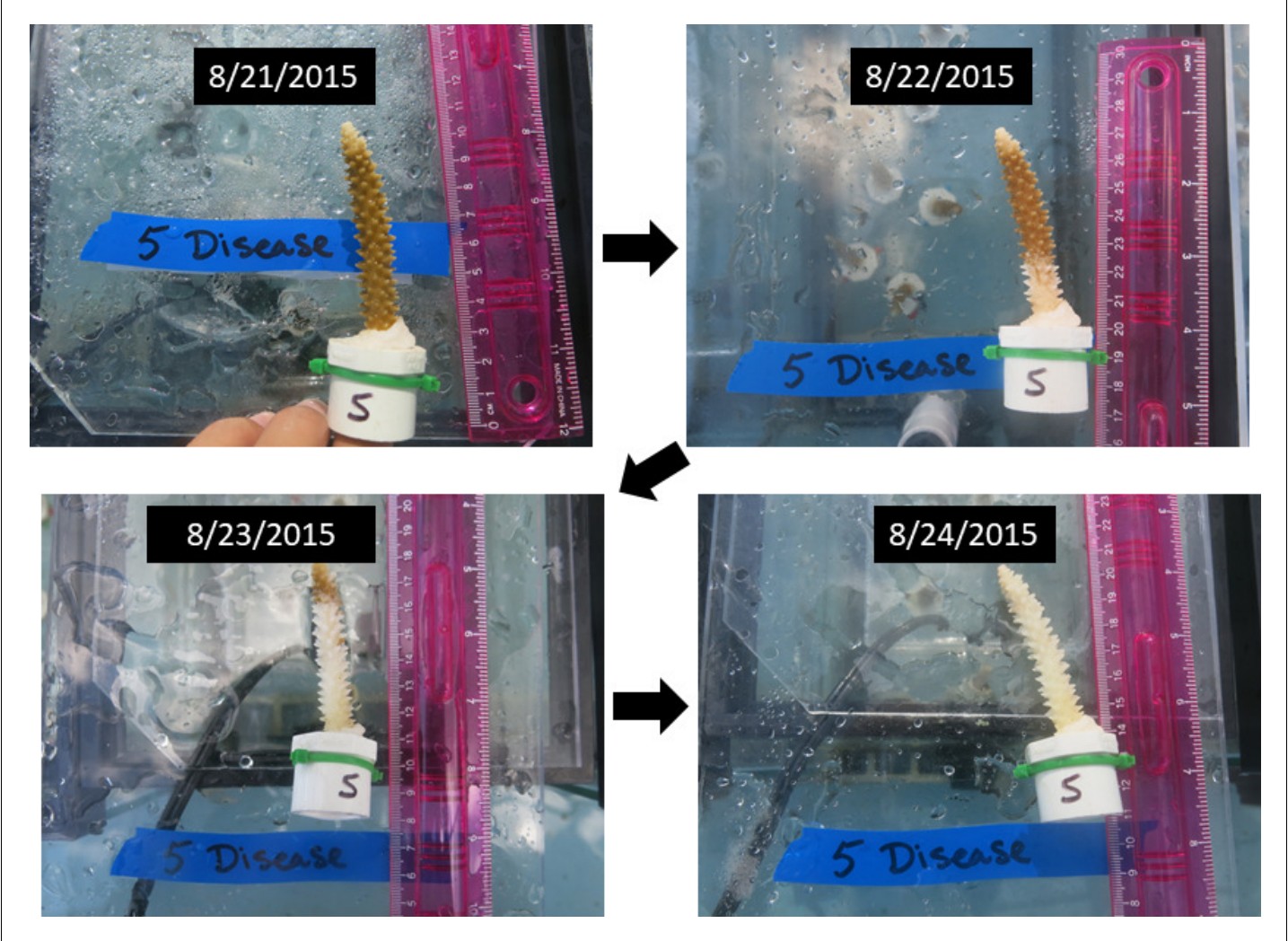

**Figure 5.** Digital photograph of an *Acropora cervicornis* fragment showing typical signs of white-band disease spreading from the base of the skeleton to the branch tip.

DOI: https://doi.org/10.7554/eLife.35066.011

hoc was used to test for differences in $F_v/F_m$ among coral genets pre- and post-bleaching because the data were not normally distributed. A Two Sample Welch's T test was used to determine whether corals with single or diverse strains of *Symbiodinium fitti* showed differing levels of photochemical efficiency pre- and post-bleaching or differed in the change of their photochemical efficiency through time. The Mann-Whitney-Wilcoxon Rank test was used to determine whether corals with single or diverse strains of *S. fitti* showed differing levels of disease susceptibility either pre or post-bleaching. A binomial generalized linear model within the 'lme4' package (*Bates et al., 2015*) was used to test whether the $F_v/F_m$ or average change in $F_v/F_m$ for each genotype between August and September, and the interaction of these factors with treatment, influenced the presence or absence of disease on each coral fragment. A binomial generalized mixed-effect linear model within the 'lme4' package (*Bates et al., 2015*) was used to test whether the fixed effects of treatment (disease vs control homogenate), genotype, and the interaction of two variables significantly influenced the presence or absence of disease manifestation within each replicate fragment. The pre-bleaching and post-bleaching trials were analyzed separately. Trial was added to the pre-bleaching analysis to determine whether there were differences in disease susceptibility among the three trials in August. Tank was identified as the random effect within the model for both the pre- and post-bleaching experiments.

## Relative risk analysis

A relative risk analysis compares the likelihood of an event occurring between two groups, individuals exposed to a risk factor versus individuals not exposed to a risk factor. Within an epidemiological setting, this analysis incorporates disease within non-exposed individuals thus accounting for chance occurrence. Traditionally, this analysis does not test for statistical significance, however, estimating the relative risk ratio within a Bayesian setting allows for statistical inference from interpretation of the posterior distribution and a comparison of results among genotypes. Within the present study, the relative risk of each genet was calculated as the number of ramets within each genet with disease after exposure to the risk (disease homogenate) divided by the number of ramets within each genet with disease that had not been exposed to the risk (healthy homogenate):

Relative risk (RR) = $\frac{Risk\ in\ exposed}{Risk\ in\ non-exposed}$, where the *risk in exposed* individuals was calculated as the incidence (diseased/total population) of those exposed to the risk and the *risk in non-exposed* individuals was calculated as the incidence of those not exposed to the risk. When RR=1 then there is no association between the exposure and disease occurrence. However when RR>1 then there is a positive association and when RR<1 there is a negative association. The posterior distribution of the relative risk was calculated using a Bayesian approach (*Gelman et al., 2004*; *Lawson, 2009*) and estimated using a binomial likelihood distribution and a uniform-Beta prior distribution. To obtain an estimate of relative risk, Markov Chain Monte Carlo simulations were used with Gibbs sampling in OpenBUGS (MRC Biostatistics Unit, Cambridge, UK, *Supplementary file 7*). Ninety-five percent credible intervals were calculated for each estimate of relative risk. Credible intervals that did not include a value of one were considered significant, with a credible interval above one signifying a higher risk of disease because of exposure to the disease homogenate. A credible interval below one signified a higher risk of disease from the lack of exposure.

Two different relative risk analyses were conducted using the present studies' data set. The initial relative risk analysis compared the prevalence of disease signs for each genet when exposed to the disease homogenate with those that were exposed to the healthy homogenate prior to bleaching (non-stressful conditions) from the August 2015 data set. The second analysis compared the prevalence of disease signs for each genet when exposed to the disease homogenate with those that were exposed to the healthy homogenate under bleached conditions from the September 2015 data set.

## Genotyping of host and symbionts

Host genotype was characterized using four host (diploid) microsatellite markers following (*Baums et al., 2005*); (*Supplementary file 8*). Previous work showed that the *A. cervicornis* corals within the nursery were dominated by *Symbiodinium 'fitti'*; no other clades were detected above background level (ca < 1%, 40). Therefore, each host genet was also sampled twice in August 2015 to determine the multi-locus genotype of the dominant dinoflagellate species, *S. 'fitti'*, using 13 algal (haploid) microsatellite markers following (*Baums et al., 2014*). Samples that returned identical multilocus algal genotypes at all loci were considered to belong to the same strain. Multilocus genotypes generated here were added to a database containing 1668 *A. cervicornis* genets and 345 *Symbiodinium fitti* strains from across the Caribbean. The probability of identity for the host is $10^{-4}$ (calculated by GenAlEx 6.503, 64) and for the symbiont is $10^{-5}$ (calculated after 63) among all samples of the respective species in the database.

## Analysis of the bacterial community within the disease and healthy homogenates

The four disease and four healthy homogenate samples were processed for next generation sequencing analysis of the bacterial community. Total DNA was extracted from each homogenate sample using the MoBio Powersoil DNA isolation kit with an extended bead-beating time of one hour (MoBio Inc., Carlsbad, CA). The bacterial community of each sample was analyzed using 16S rDNA Illumina sequencing on the MiSeq platform (see supplemental material for detailed protocol). Paired-end sequencing was performed at MR DNA (www.mrdnalab.com, Shallowater, TX, USA) using a single flow cell on a MiSeq following the manufacturer's guidelines. Sequence length averaged 450 base pairs. Sequence data were processed using MR DNA analysis pipeline (MR DNA, Shallowater, TX, USA). Sequences were joined and then depleted of barcodes. Sequences < 150 bp

and sequences with ambiguous base calls were removed. Sequences were then denoised, operational taxonomic units (OTUs) were generated and chimeras were then removed. OTUs were defined by clustering at 3% divergence (97% similarity). Final OTUs were taxonomically classified using BLASTn against a curated database derived from NCBI (www.ncbi.nlm.nih.gov) and defined based on the homology identified in *Supplementary file 9*. Illumina sequencing resulted in an average of 82,246 (± 11,393 SE) sequence reads per sample and a total of 1257 distinct operational taxonomic units (OTUs). The minimum number of reads within a sample was 39,810 and maximum reads reached 134,396. To maintain comparability among samples throughout the statistical analyses, a random subset of the minimum value, 39,810 reads, was taken from each sample prior to statistical processing.

The percent composition of bacterial groups from each sample was analyzed at the OTU level using a factorial permutation multivariate analysis of variance (PERMANOVA) with trial and homogenate type (healthy or disease) as two independent variables using the 'vegan' package of the statistical program R (*R Foundation for Statistical Computing, Vienna A, 2011*; *Oksanen, 2013*). A similarity percentages (SIMPER) analysis within the 'vegan' package (*Oksanen, 2013*) provided the percent dissimilarity between the disease and healthy homogenate caused by each bacterial OTU. The relative abundance of each OTU and each bacterial class was tested for differences among homogenate types using a Kruskal Wallis test. Bacterial OTU data were then processed through non-metric multidimensional scaling (nMDS), which applied the rank orders of data to represent the position of communities in multidimensional space using a reduced number of dimensions. The nMDS results were then plotted in two-dimensional ordination space. The average relative abundance of each bacterial class was also plotted for visualization. The sequencing data are available from GenBank within the National Center for Biotechnology Information (http://www.ncbi.nlm.nih.gov) under Accession numbers MG488295 – MG489819 for 16S rRNA gene Illumina sequencing.

## Acknowledgements

We would like to thank Meghann Devlin-Durante, Katelyn Shockman, Sara Williams, Cory Walter, Shelby Hammett, Talley Hite, Nicholas Macknight, Sammi Buckley, Mark Knowles, and David Vaughan for assistance in sample collections, data collection, and experimental implementation. We would also like to thank Caitlin Lustic for authorizing the use of nursery grown staghorn corals under permit #FKNMS-2011–150-A3. Wild coral collections for this experiment were authorized by the Florida Keys National Marine Sanctuary under permit #FKNMS-2015–084. This research was funded by NSF CAREER Award OCE-1452538 to EM and OCE-1537959 to IB.

## Additional information

### Funding

| Funder | Grant reference number | Author |
|---|---|---|
| National Science Foundation | OCE-1452538 | Erinn M Muller |
| National Science Foundation | OCE-1537959 | Iliana B Baums |

The funders had no role in study design, data collection and interpretation, or the decision to submit the work for publication.

### Author contributions

Erinn M Muller, Conceptualization, Resources, Data curation, Formal analysis, Supervision, Funding acquisition, Validation, Investigation, Visualization, Methodology, Writing—original draft, Project administration, Writing—review and editing; Erich Bartels, Conceptualization, Resources, Methodology, Writing—review and editing; Iliana B Baums, Conceptualization, Resources, Data curation, Formal analysis, Supervision, Validation, Investigation, Visualization, Methodology, Writing—review and editing

## Author ORCIDs
Erinn M Muller [iD] http://orcid.org/0000-0002-2695-2064
Iliana B Baums [iD] https://orcid.org/0000-0001-6463-7308

## Decision letter and Author response
Decision letter https://doi.org/10.7554/eLife.35066.027
Author response https://doi.org/10.7554/eLife.35066.028

## Additional files

### Supplementary files

• Supplementary file 1. Results of the two-sample t-tests and Kruskal-Wallis tests comparing the photochemical yield of 15 different genotypes of *Acropora cervicornis* prior to bleaching in August 2015 and after bleaching occurred in September 2015.
DOI: https://doi.org/10.7554/eLife.35066.012

• Supplementary file 2. *Symbiodinium fitti* multilocus genotypes. Replicate samples of the same host genet contained the same *S. fitti* genotype in all cases. Host genets 3, 4, 5, 7, 9, 10, 13, 41, 44, 47 and 58 harbor *S. fitti* strain F421.
DOI: https://doi.org/10.7554/eLife.35066.013

• Supplementary file 3. Results of the Bayesian relative risk analysis on the log scale when disease prevalence of corals exposed to the disease homogenate were compared with those that were exposed to the healthy homogenate prior to bleaching. Bold genotypes represent those that showed a significant increased risk after exposure to disease.
DOI: https://doi.org/10.7554/eLife.35066.014

• Supplementary file 4. Results of the Bayesian relative risk analysis on the log scale when disease prevalence of corals exposed to the disease homogenate were compared with those that were exposed to the healthy homogenate post bleaching. Bold genotypes represent those that showed a significant increased risk after exposure to disease.
DOI: https://doi.org/10.7554/eLife.35066.015

• Supplementary file 5. Collection information including date of collection, habitat type of collection site, and collection location in latitude and longitude of each coral genotype used within the present study. After corals were collected they were maintained and propagated within Mote Marine Laboratory's offshore *in situ* coral nursery.
DOI: https://doi.org/10.7554/eLife.35066.016

• Supplementary file 6. Experimental design and results of the four different trials used to quantify relative risk of disease for 15 different genotypes of *Acropora cervicornis*
DOI: https://doi.org/10.7554/eLife.35066.017

• Supplementary file 7: OpenBUGS code for relative risk analysis.
DOI: https://doi.org/10.7554/eLife.35066.018

• Supplementary file 8. *Acropora cervicornis* multilocus genotypes using 4 previously published microsatellite markers (*Baums et al., 2005*).
DOI: https://doi.org/10.7554/eLife.35066.019

• Supplementary file 9. Standardized grouping information to identify each operational taxonomic unit to the most accurate taxonomic level.
DOI: https://doi.org/10.7554/eLife.35066.020

• Transparent reporting form
DOI: https://doi.org/10.7554/eLife.35066.021

### Data availability

The sequencing data are available from GenBank within the National Center for Biotechnology Information (http://www.ncbi.nlm.nih.gov) under accession numbers MG488295 - MG489819 for 16S rRNA gene Illumina sequencing.

The following datasets were generated:

| Author(s) | Year | Dataset title | Dataset URL | Database, license, and accessibility information |
|---|---|---|---|---|
| Muller, Erinn | 2015 | Disease Exposure Experiment | http://www.bco-dmo.org/dataset/642860 | Publicly available (accession no.) 642860 |
| Erinn M Muller | 2017 | Uncultured bacterium 16S ribosomal RNA gene, partial sequence. | https://www.ncbi.nlm.nih.gov/popset?LinkName=nuccore_popset&from_uid=1276596883 | Publicly available at the NCBI GenBank (accession no. MG488295 - MG489819) |

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
