## [Decision Letter]

Thank you for submitting your article "Bleaching causes loss of disease resistance within the threatened coral species, *Acropora cervicornis*" for consideration by *eLife*. Your article has been reviewed by three peer reviewers, and the evaluation has been overseen by a Reviewing Editor and Ian Baldwin as the Senior Editor. The following individuals involved in review of your submission have agreed to reveal their identity: Noah Rose (Reviewer #1); Dave Baker (Reviewer #3).

The Reviewing Editor has drafted this decision to help you prepare a revised submission.

Summary:

The data here arose from an opportunity to contrast disease susceptibility in a lab population of corals before and after a bout of warm water exposure. They found disease to be more prevalent after infection in September than in August: about 74% among clones in September and about 28% (averaged across trials) in August. One especially good aspect of the study was the attention to individual clones. In particular, four coral clones showed no disease. There was no relationship between loss of algal photosynthetic ability (a proxy for bleaching) and disease susceptibility. The data set is unique and of strong interest. It is based on careful measurement of disease and photosynthesis, among coral colonies, and contributes strongly in a number of important ways.

The simplicity and clarity of the data set is a little muddled by some parts of the methods and analysis and some ways the data are presented. For example, the basic experiment is uncontrolled. It compares an experiment done in August to one done in September. In each month, a slurry of infectious stuff is created and applied, but there is no way of standardizing the slurry, or knowing if the slurry was more or less powerful in August than September. The disease incidence among the controls is much higher in September than August (17% versus 1%), with half the signal coming from four clones (4, 13, 47, 57). This shows that disease susceptibility increases without experimental exposure, a result that supports the finding of a similar trend among the treated samples (28% to 73% shift in disease). Without these control colonies, a conclusion that disease susceptibility increased in September couldn't be supported because the application of the disease agent was not standardized.

The most important result is the finding of differences in disease susceptibility among clones all treated with the same slurry. This part of the experiment does not suffer from the lack of controls cited above because it is a comparison among samples all treated the same way. Here, there are data on relative disease among clones pre-bleaching, and post-bleaching and in controls. And there are data on a proxy for bleaching. None of these measures look correlated among the 15 clones, though the manuscript does not actually do an analysis of this.

A second aspect that will need clarification is some discussion about why bleaching was not measured. The F_v_/F_m_ measure is a good proxy for photosynthetic damage that often leads to bleaching. But it is not bleaching. Did the corals bleach? If the corals did not actually lose pigment, then this seems like an important aspect of the results. If they did bleach, a visual bleaching score, or even a statement that the corals bleached in concert with the F_v_/F_m_, seems needed to link the data you collected to the phenomenon you focus on. Barring that, there needs to be a simple statement that you measured a common proxy of bleaching, not bleaching itself.

A third aspect is that you focus on the temperature differences between August and September, but because this is an uncontrolled experiment, there could be many things that changed in the interim: microbiome, oxygen, pH, sediment, parasites, etc. This does not impact the comparison among clones within time points, of course. Some caution about the cause of differences in disease from August to September seems reasonable given the way the two tests were performed.

Overall, the experiment is treated as if it were a controlled test of bleaching state and disease susceptibility. In fact, it reports an uncontrolled exposure of clones to infectious agents at two time points. These are interesting and valuable experiments, but describing them as relating temperature tolerance (which wasn't measured specifically) or bleaching (which wasn't measured specifically) to disease risk (which was not standardized across dates) are inferences about the data, not really a description of the results. You'll have to do a little more work linking the data set (a nice measurement of disease at two time points) to these inferences.

The strongest part of the data set is the comparison among clones. Here there seems to be no relationship between disease and photosynthesis impairment. If there is no relationship between the two, how can death by bleaching imperil disease resistance? It seems like a selective event that leads to only the most temperature tolerant genotypes surviving would have no impact on the prevalence of disease resistance in a large population. This is because some highly disease resistant clones would die of bleaching, but so would some highly disease susceptible ones. Only if disease resistance and heat resistance were inversely correlated would selecting for heat survivors negatively impact disease resistance.

Maybe your intuition here is being driven by these being small populations? And in this case, any winnowing of the highly disease resistant clones might be a serious problem. A rethink of this might be extremely useful for the broader coral community, which is not used to thinking about population numbers of a coral species being less than condors.

Essential revisions:

– Subsection “Disease and healthy homogenates” – measure density of slurry somehow? Was the second slurry the same density of infectious microbes?

– Subsection “Photochemical efficiency” – why wasn't bleaching measured?

– Discussion section, first paragraph – "Disease resistance and temperature tolerance appear to evolve independently" *but* temperature tolerance was not measured. It is inferred.

– Discussion section, first paragraph – "driven by the host genotype" *but* no other factor was tested, you only tested host genotype and symbiont. This means that other aspects of the clones – their position in the tanks, microbiomes, size, reproductive status, color, etc. – have no chance of being significant. I don't doubt there are genotype effects, but typically an analysis would try to control for as many non-genetic aspects as possible. Some attempt to examine the data for unexpected impacts of strange lab effects seems like a good idea.

– Discussion section, first paragraph – "this represents a 2,000 times increase in disease risk" – see reviewer 1’s review. I find these risk ratios obscuring. In the trials you list in Supplementary file 6, there were 44%, 15%, and 31% disease rates among clones in August, and 73% in September. How is this 2000 times higher?

Along these lines, Figure 2 should be dropped and some other way of displaying the results should be found. As I read Figure 2, only genets 9 and 10 had significant disease risk in August. In September, clones 5, 9, 41, 44, 46, 50 had disease risk. Is that what you are trying to say?

– Discussion section, second paragraph – "… 27%, of the tested population showed complete resistance to disease exposure" – these were clones 3, 7, 41, 44. They were tested about 5 times each. Comparing these numbers to Panama and USVI requires that similar number of tests were done in each location. For example, if more ramets were tested in Panama per genet, then the incidence of complete resistance might be lower.

*Reviewer #1:*

In their manuscript, "Bleaching causes loss of disease resistance within the threatened coral species, *Acropora cervicornis*," Muller and colleagues address a problem of vital importance in contemporary evolutionary biology and conservation biology. Coral reefs around the world are greatly threatened by coral bleaching. However, bleaching itself doesn't typically kill corals. Instead, some combination of disease and starvation lead to mass die-offs across coral reefs in the weeks and months following the actual bleaching event. The likelihood of successful adaptation depends critically on the relationship between bleaching tolerance and disease tolerance. Muller et al. provide convincing evidence that different genotypes of *Acropora cervicornis* differ in both bleaching tolerance and disease tolerance, but that these two traits are uncorrelated. This is an important finding that suggests that focusing on conserving or propagating corals that are resilient with regard to just one trait, like bleaching resistance, could result in the loss of diversity in disease resistance traits that could play a key role in coral reef persistence in the coming decades and centuries. However, I have some concerns about the data analysis and some of the conclusions and interpretations drawn. These concerns would have to be addressed before this manuscript would be suitable for publication.

1) Much of the analysis rests on comparing median rates of relative risk in different disease treatments to different controls on a per-genet basis. While this Bayesian analysis does a useful job of summarizing risk rates for each genet, this analysis doesn't explicitly test whether genets differ significantly among each other in disease susceptibility either before or after bleaching. I think it is critical that this manuscript include an explicit test for genotype effects (not just genet-level tests for significant risk or lack of significant risk as presented in Supplementary files 3-4). A Chi-squared test within each treatment or a binomial generalized linear model (glm) could work.

2) I think there needs to be a greater discussion of what exactly happened to the control bleached ramets that died. Were they all in particular sea tables? If the authors saw differences in disease development between different sea tables, this needs to be accounted for in modeling. In any event, high disease mortality among bleached controls suggests that they may be bringing in pathogens and that the treatment increases the level of exposure but doesn't make the difference between exposure and non-exposure. Or if the authors believe control mortality may have some causes besides this disease, this needs to be further discussed with regard to both treatments and controls.

3) The Abstract and last paragraph of the Discussion state that bleaching increased disease-induced mortality six-fold. This statement is misleading since the six-fold (natural log scale) increase is relative to non-bleached non-exposed ramets, and so it reflects the cumulative effects of both stressors. In a back-of-the-envelope sense, 25 ramets of 75 developed disease before bleaching (33%) and 55 of 75 ramets developed disease after bleaching (72%); this is much closer to a two-fold (this is not log scale) increase in risk under bleaching conditions. If comparing resistance among bleached corals to previous levels of resistance prior to bleaching, the valid comparison is between disease susceptibility in bleached exposed and unbleached exposed, not bleached exposed and unbleached unexposed. Also, the 2000 fold increase in the first paragraph of the Discussion doesn't make sense to me. e^6^ is about 400, which still seems a bit high – once again back of the envelope estimates would suggest that the control risk is 1/75 (about 1%) and exposed bleached is 55/75 or 72%. Something seems off here. I think by taking the median risk ratio the authors are taking the ratio from a genet that never developed disease in the controls (since only one genet ever developed disease among the controls), so this risk ratio is entirely a function of the uniform prior and sample size, rather than being based on actual observations of disease among both groups. I think the cumulative effect part of the analysis should probably be replaced altogether with a comparison of risk among bleached exposed relative to unbleached exposed.

*Reviewer #2:*

The authors here report the findings of a comprehensive study of disease resistance in *Acropora palmata* collected from the Florida Keys, and the effects of thermal bleaching on this resistance. The reported study is thorough, and the reported findings are of particular interest to a diversity of coral biologists as well as those interested in marine invertebrate ecology. Generally the authors have been exceptionally thorough in their description and explanation of their findings, however there are a few aspects of the data that I would like to see addressed prior to publication.

First, while the authors explored the effects of symbiont genotype/strain on bleaching and disease tolerance, there is no report of any examination of the effects of symbiont density on these same responses. Considering other studies indicating that symbiont density does affect host bleaching susceptibility (Cunning and Baker, 2014; Cunning et al., 2015), I believe it is important for the authors to address any observed effects of variation in symbiont density (if the data is readily available).

Second, I would like to see some analysis regarding the rates of disease progression for infected corals. This is particularly of interest in the case of variable disease tolerance (i.e. did some colonies become infected, but show much slower rates of disease progression)? It would appear that the necessary data for these analyses was collected but not reported on. I believe that this data should be at least presented in supplementary materials and noted to in the main text of the manuscript. Finally, I am curious to see if the authors did any statistical analyses to determine if there were any 'batch' effects so to speak as a result of running the first experiment in three trials. Either way this should be noted in the manuscript.

Less significant items that I would also like to see addressed in the final manuscript include some kind of note on future directions in determining how resistance varies between Florida Keys and Panama populations of *A. palmata* (Discussion, second paragraph). The authors note the apparent genetic basis of shared resistance to disease and bleaching in Panamanian *A. palmata*, and that this does not appear to be the case in the Florida Keys. However they make no mention of future directions to study the genetic basis of this difference. The manuscript would be strengthened by some mention of future directions of study in this realm. Additionally I would like to see further discussion of coral genets 7 and 58, both of which show resistance under all three analyses conducted. Perhaps the authors could note suggestions for future study of these genets in particular in order to determine what makes them resistant in particular.

In summary, the submitted manuscript is a concise and thorough report of a study with significant implications for coral biology and restoration efforts. I believe with minor modifications it will be well received by a diversity of scientists.

*Reviewer #3:*

Well, this is a delightful paper to review because it's extremely well designed and edited. The question and hypothesis are novel and interesting and the results are convincing. I strongly support its publication.

The finding of no significant relationship with *Symbiodinium* strain on disease resistance is quite useful to disseminate as often the symbionts are viewed as the Achille's heel of the holobiont. There is also a very cautionary and important message about the implications for coral restoration.

One suggestion: The mechanisms posed for why some hosts are disease resistant lean towards the purely molecular/biochemical – but what if those genotypes are simply more nutritionally competent to endure bleaching and resist disease? In a simple view, any organism deficient in diet will succumb to disease. It has been shown in many prior works that corals who can supplement their diets from heterotrophy can endure bleaching events and recover more rapidly.

Following that, it seems logical to me that this paper accomplishes several things: 1) it refutes the hypothesis that *Symbiodinium* identity influences host pre- and post-bleaching disease resistance, 2) it identifies particular host genotypes that display significant resistance, despite the fact that symbiont performance is severely impaired, but then 3) it evokes gene regulation as the mechanism for explaining (2).

I think the far more parsimonious extension of their data is that those genotypes are, perhaps through a multitude of mechanisms, less metabolically active than their conspecifics. It would be worth mentioning energetics in this paper as a possible explanation for resistance (perhaps this is not mutually exclusive from impaired gene expression), and, in the future, an interesting "value add" to such studies to examine the host physiology, i.e. lipid content as a measure of energy reserves.

A sentence or two could accomplish this in a very minor revision.

An enjoyable paper to read, indeed! Great work.

---

## [Author Response]

[…] The simplicity and clarity of the data set is a little muddled by some parts of the methods and analysis and some ways the data are presented. For example, the basic experiment is uncontrolled. It compares an experiment done in August to one done in September. In each month, a slurry of infectious stuff is created and applied, but there is no way of standardizing the slurry, or knowing if the slurry was more or less powerful in August than September. The disease incidence among the controls is much higher in September than August (17% versus 1%), with half the signal coming from four clones (4, 13, 47, 57). This shows that disease susceptibility increases without experimental exposure, a result that supports the finding of a similar trend among the treated samples (28% to 73% shift in disease). Without these control colonies, a conclusion that disease susceptibility increased in September couldn't be supported because the application of the disease agent was not standardized.

Thank you very much for this thoughtful perspective. We have removed the cumulative analysis from the manuscript and now simply treat the results as two separate experiments.

The most important result is the finding of differences in disease susceptibility among clones all treated with the same slurry. This part of the experiment does not suffer from the lack of controls cited above because it is a comparison among samples all treated the same way. Here, there are data on relative disease among clones pre-bleaching, and post-bleaching and in controls. And there are data on a proxy for bleaching. None of these measures look correlated among the 15 clones, though the manuscript does not actually do an analysis of this.

We originally conducted Spearman rank correlations to determine whether there were associations between pre- and post-bleaching photochemical efficiency and pre- and post-bleaching disease susceptibility. The change in photochemical efficiency of each genotype was also tested for associations with pre- and post-bleaching disease susceptibility using a Spearman rank correlation. Within the first draft, we reported in the Results that none of the analyses were statistically significant, suggesting that bleaching severity (measured with F_v_/F_m_ as a proxy) was not correlated with disease susceptibility. Within the present revision, we analyzed the data using a binomial generalized linear model and again show that F_v_/F_m_ does not affect the presence or absence of bleaching, which is a much more parsimonious analysis.

A second aspect that will need clarification is some discussion about why bleaching was not measured. The F_v_/F_m_ measure is a good proxy for photosynthetic damage that often leads to bleaching. But it is not bleaching. Did the corals bleach? If the corals did not actually lose pigment, then this seems like an important aspect of the results. If they did bleach, a visual bleaching score, or even a statement that the corals bleached in concert with the F_v_/F_m_, seems needed to link the data you collected to the phenomenon you focus on. Barring that, there needs to be a simple statement that you measured a common proxy of bleaching, not bleaching itself.

Bleaching was not scored qualitatively, but F_v_/F_m_ was used as a proxy. Visually, it was obvious that all corals bleached between the August and September trials. We have added an August and September photo of each genotype for visualization of the actual bleaching that took place. We have also added statements within the Materials and methods that discusses F_v_/F_m_ as a proxy and reports that all corals were visually bleached in September, but visually pigmented in August.

A third aspect is that you focus on the temperature differences between August and September, but because this is an uncontrolled experiment, there could be many things that changed in the interim: microbiome, oxygen, pH, sediment, parasites, etc. This does not impact the comparison among clones within time points, of course. Some caution about the cause of differences in disease from August to September seems reasonable given the way the two tests were performed.

You are quite right. We have removed the cumulative analysis, which compared the two experiments and focused on within experiment results.

Overall, the experiment is treated as if it were a controlled test of bleaching state and disease susceptibility. In fact, it reports an uncontrolled exposure of clones to infectious agents at two time points. These are interesting and valuable experiments, but describing them as relating temperature tolerance (which wasn't measured specifically) or bleaching (which wasn't measured specifically) to disease risk (which was not standardized across dates) are inferences about the data, not really a description of the results. You'll have to do a little more work linking the data set (a nice measurement of disease at two time points) to these inferences.

We have reworked the Discussion to reflect the independent results of the two experiments. We include discussion of the reasons for the potential differences, and use caution when comparing the results of these two independent studies through added text that addresses the difficulty in determining causation of results because the pathogenic load and virulence could not be assessed.

The strongest part of the data set is the comparison among clones. Here there seems to be no relationship between disease and photosynthesis impairment. If there is no relationship between the two, how can death by bleaching imperil disease resistance? It seems like a selective event that leads to only the most temperature tolerant genotypes surviving would have no impact on the prevalence of disease resistance in a large population. This is because some highly disease resistant clones would die of bleaching, but so would some highly disease susceptible ones. Only if disease resistance and heat resistance were inversely correlated would selecting for heat survivors negatively impact disease resistance.

Another good point. Thank you. We have now readdressed the results to conclude that tolerance to heat stress may not confer resistance to disease in *A. cervicornis*. These traits appear to be independent from each other, and efforts to include disease resistance in concert with heat tolerance will be essential, if selective breeding is a tool being incorporated into restoration.

Maybe your intuition here is being driven by these being small populations? And in this case, any winnowing of the highly disease resistant clones might be a serious problem. A rethink of this might be extremely useful for the broader coral community, which is not used to thinking about population numbers of a coral species being less than condors.

We have better contextualized our conclusions, which have also been adjusted to reflect that these traits appear independent from each other and the loss of any corals through bleaching may winnow disease resistance inadvertently. Indeed, we are working with currently small population sizes in the Florida Keys and sexual recruitment is very low. Together, this increases the concern over losing disease resistant clones to bleaching.

Essential revisions:– Subsection “Disease and healthy homogenates” – measure density of slurry somehow? Was the second slurry the same density of infectious microbes?

Unfortunately, there was no way to determine the density of infectious microbes as the pathogen responsible for white band disease is unknown. However, we added a paragraph within the Materials and methods that outlines the standardization procedures put into place in an attempt to keep these two experiments comparable. We also added in the microbiome analysis of the homogenates, which showed that the bacterial community did not significantly differ among trials. Additionally, we removed the cumulative risk analysis which attempted to compare the results of the two experiments. In the revised manuscript, we limit our comparisons between the two experiments and focus on the results within each experiment.

– Subsection “Photochemical efficiency” – why wasn't bleaching measured?

To aid in the reader’s interpretation of the bleaching state of the corals, we have added visual representations of each genotype within the pre-bleaching and post bleaching experiments. There were such strong visual changes that the qualitative results would show ‘normal pigmentation’ in August and ‘completely bleached tissue’ in September. The PAM F_v_/F_m_ values were used to better quantify the extent of bleaching and could be utilized for more robust statistical analyses.

*– Discussion section, first paragraph – "Disease resistance and temperature tolerance appear to evolve independently"* but *temperature tolerance was not measured. It is inferred.*

Although F_v_/F_m_ is a proxy for bleaching, the strong visual change within each coral genotype supports the results that bleaching occurred, and was not just inferred. The F_v_/F_m_ values were measured to better quantify the photochemical efficiency of algal symbionts that remained within the corals after pigmentation could not be observed with the naked eye. Because of this, it provides a better quantification of the bleached state within each coral.

*– Discussion section, first paragraph – "driven by the host genotype"* but *no other factor was tested – you only tested host genotype and symbiont. This means that other aspects of the clones – their position in the tanks, microbiomes, size, reproductive status, color, etc. – have no chance of being significant. I don't doubt there are genotype effects, but typically an analysis would try to control for as many non-genetic aspects as possible. Some attempt to examine the data for unexpected impacts of strange lab effects seems like a good idea.*

We have analyzed the data to determine whether treatment (exposure to disease homogenate or control), host genotype, symbiont type, photochemical efficiency of the symbiont, tank, and trial influenced the presence or absence of disease occurrence on each replicate coral. Coral position was random within the tank. The only significant fact was treatment. However, the relative risk analysis shows that genotypes respond differently to exposure to the disease. We discuss this in more detail within the discussion and include the possibility that several mechanisms may be influencing this conclusion, including the host genome, microbiome, and energetics with the host.

– Discussion section, first paragraph – "this represents a 2,000 times increase in disease risk" – see reviewer 1’s review. I find these risk ratios obscuring. In the trials you list in Supplementary file 6, there were 44%, 15%, and 31% disease rates among clones in August, and 73% in September. How is this 2000 times higher?

We agree that these conclusions are confusing and have removed them from the manuscript.

Along these lines, Figure 2 should be dropped and some other way of displaying the results should be found. As I read Figure 2, only genets 9 and 10 had significant disease risk in August. In September, clones 5, 9, 41, 44, 46, 50 had disease risk. Is that what you are trying to say?

Yes. Your interpretation of the figures is correct and the text within the Results state these very conclusions. To ease in the interpretation of the results, we have removed the cumulative risk analysis, and separated out the two relative risk analyses to be presented independently (now Figure 3A and 3B), which better represents the manuscripts two independent studies and eases interpretation.

– Discussion section, second paragraph – "… 27%, of the tested population showed complete resistance to disease exposure" – these were clones 3, 7, 41, 44. They were tested about 5 times each. Comparing these numbers to Panama and USVI requires that similar number of tests were done in each location. For example, if more ramets were tested in Panama per genet, then the incidence of complete resistance might be lower.

The same number of replicates (n=5) was assessed in the Panama study, however, the VI study was a field observation study within variability among replicate genotypes, but were monitored for almost ten years to conclude disease resistance. Although the methods differed in the VI study, the comparison is still included with a statement that includes there are differences in methods among studies.

Reviewer #1:[…] 1) Much of the analysis rests on comparing median rates of relative risk in different disease treatments to different controls on a per-genet basis. While this Bayesian analysis does a useful job of summarizing risk rates for each genet, this analysis doesn't explicitly test whether genets differ significantly among each other in disease susceptibility either before or after bleaching. I think it is critical that this manuscript include an explicit test for genotype effects (not just genet-level tests for significant risk or lack of significant risk as presented in Supplementary files 3-4). A Chi-squared test within each treatment or a binomial generalized linear model (glm) could work.

Thank you very much for this great suggestion. We have added in a binomial generalized linear model with all of the potential covariates to test which effects influence disease infection. Within this analysis we found that only treatment (exposure to disease) influenced disease presence or absence.

2) I think there needs to be a greater discussion of what exactly happened to the control bleached ramets that died. Were they all in particular sea tables? If the authors saw differences in disease development between different sea tables, this needs to be accounted for in modeling. In any event, high disease mortality among bleached controls suggests that they may be bringing in pathogens and that the treatment increases the level of exposure but doesn't make the difference between exposure and non-exposure. Or if the authors believe control mortality may have some causes besides this disease, this needs to be further discussed with regard to both treatments and controls.

Each trial occurred within only one raceway at a time, as each raceway can hold ten different 5 gallon tanks (n=5 disease tanks, and n=5 control tanks). So, there was no variation in raceways, but we did explore tank effect, which was added with the generalized linear model analysis as a random factor. We did not find statistical differences among tanks within the same treatment, only treatment effects.

3) The Abstract and last paragraph of the Discussion state that bleaching increased disease-induced mortality six-fold. This statement is misleading since the six-fold (natural log scale) increase is relative to non-bleached non-exposed ramets, and so it reflects the cumulative effects of both stressors. In a back-of-the-envelope sense, 25 ramets of 75 developed disease before bleaching (33%) and 55 of 75 ramets developed disease after bleaching (72%); this is much closer to a two-fold (this is not log scale) increase in risk under bleaching conditions. If comparing resistance among bleached corals to previous levels of resistance prior to bleaching, the valid comparison is between disease susceptibility in bleached exposed and unbleached exposed, not bleached exposed and unbleached unexposed.

We have removed the cumulative risk analysis from the manuscript. Ideally, we would be able to complete the full factorial design within a single study, which is in our scope for future work (see next comment response).

Also, the 2000 fold increase in the first paragraph of the Discussion doesn't make sense to me. e^6^ is about 400, which still seems a bit high – once again back of the envelope estimates would suggest that the control risk is 1/75 (about 1%) and exposed bleached is 55/75 or 72%. Something seems off here. I think by taking the median risk ratio the authors are taking the ratio from a genet that never developed disease in the controls (since only one genet ever developed disease among the controls), so this risk ratio is entirely a function of the uniform prior and sample size, rather than being based on actual observations of disease among both groups. I think the cumulative effect part of the analysis should probably be replaced altogether with a comparison of risk among bleached exposed relative to unbleached exposed.

You are quite right that this text is confounded and confusing. We have removed this section as well as the cumulative analysis. We were in the process of completing the trifecta study of high temperature, ocean acidification, and disease on 20 different genotypes when Hurricane Irma hit our lab and destroyed the experiment in 2017. We plan on tackling this cumulative effect again in 2019, keeping our fingers crossed for no hurricanes that year.

Reviewer #2:[…] First, while the authors explored the effects of symbiont genotype/strain on bleaching and disease tolerance, there is no report of any examination of the effects of symbiont density on these same responses. Considering other studies indicating that symbiont density does affect host bleaching susceptibility (Cunning and Baker, 2014; Cunning et al., 2015), I believe it is important for the authors to address any observed effects of variation in symbiont density (if the data is readily available).

Although destructive sampling was considered in order to assess parameters such as symbiont density, we ultimately decided to retain surviving corals for subsequent student studies. Growing and maintain corals for restoration is time consuming and costly and having nursery reared corals for research is a huge advantage. Instead of destroying all corals at the end of the study for subsequent testing, having the ability to further use the corals for non-destructive studies seemed like the better choice in the long run. This decision is even more critical now as there are no corals available for research purposes after hurricane Irma destroyed many coral nurseries within the lower Keys region. Having living corals on hand during this time allows small research projects to persist, when otherwise we would be hard pressed to have living organisms to study in a laboratory setting.

Second, I would like to see some analysis regarding the rates of disease progression for infected corals. This is particularly of interest in the case of variable disease tolerance (i.e. did some colonies become infected, but show much slower rates of disease progression)? It would appear that the necessary data for these analyses was collected but not reported on. I believe that this data should be at least presented in supplementary materials and noted to in the main text of the manuscript.

Obtaining disease progression rates was originally a goal of the pre- and post-bleaching experiments. However, the data produced is not robust enough to determine these rates accurately. We have good data for corals that showed slow rates of progression through an entire day or two, but at times the entire coral fragment would lose all tissue overnight. The ~15 hour time frame between dusk (last observation) and the first measurement was too long of a time frame to actually gather rate information. Additionally, some genets showed only one fragment with disease signs during the study, which limits comparability among genets (n=1 compared with n=5 at times). Finally, disease on bleached corals manifested all at once overnight so that we would wake up to find the coral completely dead. Because this happened for every genet with disease when bleached, comparing this data would provide no information on differential genet susceptibility within studies.

Finally, I am curious to see if the authors did any statistical analyses to determine if there were any 'batch' effects so to speak as a result of running the first experiment in three trials. Either way this should be noted in the manuscript.

Within the statistical analyses we have added a fixed factor of ‘Trial’, which was not statistically significant. Additionally, we added in the 16S microbiome data, which characterized the bacterial communities for each control and disease homogenate used within the study. Again, we did not see differences among Trials.

Less significant items that I would also like to see addressed in the final manuscript include some kind of note on future directions in determining how resistance varies between Florida Keys and Panama populations of A. palmata (Discussion, second paragraph). The authors note the apparent genetic basis of share resistance to disease and bleaching in Panamanian A. palmata, and that this does not appear to be the case in the Florida Keys. However they make no mention of future directions to study the genetic basis of this difference. The manuscript would be strengthened by some mention of future directions of study in this realm.

At the conclusion of the paragraph we state “Future work should concentrate on determining the degree of spatial variability among temperature and infectious disease resistance traits and their interactions.”

Additionally I would like to see further discussion of coral genets 7 and 58, both of which show resistance under all three analyses conducted. Perhaps the authors could note suggestions for future study of these genets in particular in order to determine what makes them resistant in particular.

We have now included discussion of the possibilities conferring disease resistance within the genotypes that showed disease resistance, primarily genotypes 3 and 7. Thank you for this suggestion.

In summary, the submitted manuscript is a concise and thorough report of a study with significant implications for coral biology and restoration efforts. I believe with minor modifications it will be well received by a diversity of scientists.

Thank you very much for your time, consideration, and insight into this review. Your contributions have made this a much stronger paper, which will also be a great contribution to reef science because of your review.

Reviewer #3:[…] One suggestion: The mechanisms posed for why some hosts are disease resistant lean towards the purely molecular/biochemical – but what if those genotypes are simply more nutritionally competent to endure bleaching and resist disease? In a simple view, any organism deficient in diet will succumb to disease. It has been shown in many prior works that corals who can supplement their diets from heterotrophy can endure bleaching events and recover more rapidly.

Excellent point. We have now added significant discussion to postulate the many different mechanisms that could be driving disease resistance including molecular pathways, microbial influences, and energetics.

Following that, it seems logical to me that this paper accomplishes several things: 1) it refutes the hypothesis that Symbiodinium identity influences host pre- and post-bleaching disease resistance, 2) it identifies particular host genotypes that display significant resistance, despite the fact that symbiont performance is severely impaired, but then 3) it evokes gene regulation as the mechanism for explaining (2).I think the far more parsimonious extension of their data is that those genotypes are, perhaps through a multitude of mechanisms, less metabolically active than their conspecifics. It would be worth mentioning energetics in this paper as a possible explanation for resistance (perhaps this is not mutually exclusive from impaired gene expression), and, in the future, an interesting "value add" to such studies to examine the host physiology, i.e. lipid content as a measure of energy reserves.A sentence or two could accomplish this in a very minor revision.

Thank you very much for this insightful addition to the paper. We have added a couple of sentences that discuss the possibility that disease resistant corals may also have different energetic demands or metabolic levels compared with disease susceptible corals.